# Malignant Pleural Mesothelioma: Preliminary Toxicity Results of Adjuvant Radiotherapy Hypofractionation in a Prospective Trial (MESO-RT)

**DOI:** 10.3390/cancers15041057

**Published:** 2023-02-07

**Authors:** Elisabetta Parisi, Donatella Arpa, Giulia Ghigi, Lucia Fabbri, Flavia Foca, Luca Tontini, Elisa Neri, Martina Pieri, Simona Cima, Marco Angelo Burgio, Maria Luisa Belli, Luca Luzzi, Antonino Romeo

**Affiliations:** 1Radiotherapy Unit, IRCCS Istituto Romagnolo per lo Studio dei Tumori (IRST) “Dino Amadori”, 47014 Meldola, Italy; 2Unit of Biostatistics and Clinical Trials, IRCCS Istituto Romagnolo per lo Studio dei Tumori (IRST) “Dino Amadori”, 47014 Meldola, Italy; 3Department of Medical Oncology, IRCCS Istituto Romagnolo per lo Studio dei Tumori (IRST) “Dino Amadori”, 47014 Meldola, Italy; 4Medical Physics Unit, IRCCS Istituto Romagnolo per lo Studio dei Tumori (IRST) “Dino Amadori”, 47014 Meldola, Italy; 5Department of Medical, Surgical and Neuroscience Sciences, University of Siena, 53100 Siena, Italy

**Keywords:** mesothelioma, pulmonary cancer, radiotherapy, tomotherapy, toxicity, hypofractionation, radiation pneumonitis, OS

## Abstract

**Simple Summary:**

In this publication, we report preliminary toxicity results of a prospective phase II trial where the first 20 patients with in lung presence of pleural mesothelioma were treated with accelerated hypofractionated radiotherapy. No G3-G4 acute and late toxicity was found, while the most common acute toxicity was pneumonitis with 65.0% G1 and 10% G2. The median OS was 33.1 months (95% CI:14.4–not estimable) and the median Time To Progression was 18.2 months (95% CI:11.3–not estimable). The trial is ongoing, but these results can be considered encouraging.

**Abstract:**

Malignant Pleural Mesothelioma (MPM) is a rare malignancy with an overall poor prognosis. The standard therapeutic strategy in early-stage disease is trimodality therapy. In this publication, we report the preliminary toxicity results of the first 20 patients treated with accelerated hypofractionated radiotherapy. Between July 2017 to June 2019, 20 MPM patients were enrolled and treated with accelerated hypofractionated radiotherapy using helical tomotherapy and intensity-modulated arc therapy. The prescription dose was 30 Gy in five daily fractions, while an inhomogeneous dose escalation to 40 Gy was prescribed based solely upon the presence of gross residual tumor. Only one case of G3 toxicity was reported, which was a bilateral pneumonitis that occurred two years after treatment probably due to superinfection. Median Time to Progression reached 18.2 months while one- and three-year Overall Survival rates were 85% (95% CI:60.4–94.9) and 49.5% (95% CI:26.5–68.9), respectively. Treatment of the intact lung with pleural intensity-modulated arc irradiation is a novel treatment strategy that appears to be safe, feasible, and without a high grade of lung toxicity. Survival rates and Time to Progression are encouraging.

## 1. Introduction

Malignant pleural mesothelioma (MPM) is a relatively rare aggressive malignancy [1] of the pleural caused by prior asbestos exposure. It has a poor prognosis, with a median survival time of less than 12 months after diagnosis [2,3]. Treatment failure is most common in the ipsilateral hemithorax, so optimizing local control provides the best opportunity for long-term survival. In the past, treatment options were platinum-based chemotherapy and demonstrated survival benefits in randomized trials [2]. Regarding the surgical approach, two principal surgeries are performed for mesothelioma: extrapleural pneumonectomy (EPP) and pleurectomy/decortication (P/D) [4,5,6]. Historically, radiotherapy had two different roles in the treatment of MPM: a palliative role effective at controlling pain in some patients [7] and a prophylactic role to prevent the development of cutaneous metastases at the sites of previous pleural interventions. Recently, radiotherapy has been employed with an adjuvant intent to the hemithorax in the context of trimodality treatment, especially after EPP. The addition of postoperative chest cavity radiotherapy has been shown to improve local control and survival [8]. Technically, the use of Intensity-modulated radiation therapy (IMRT) [9,10], and in particular, intensity-modulated arc therapy (IMAT) would appear to be the most effective for adjuvant treatment in homolateral lung presence [11,12]. Considering the high rate of intraoperative mortality and morbidity for EPP, as reported in the MARS trial [13], the practice of extrapleural pneumonectomy has waned, and the practice of pleurectomy/decortication has increased. The aim of this surgical approach is only tumoral cytoreduction and complete macroscopic resection. In this scenario, the role of radiation therapy remains to be defined concerning the potential risk of lung toxicity. There is a similar scenario for patients who cannot undergo surgery but who do not complain of symptoms. Effectively, for these patient settings, radiotherapy is a challenge because of the risk of high-grade pneumonitis in the intact lung. Only mono-institutional experiences are reported in the literature with evidence of IMRT and conventional fractionation after P/D or biopsy. These experiences underline the feasibility and acceptable toxicity profile of the treatment [14,15]. To date, no large-scale studies have been found in the literature regarding alternatives to the conventional dose schemes in the adjuvant setting such as accelerated hypofractionated radiotherapy, while SBRT schemes were explored in operated patients, but only as salvage therapy for oligorecurrent and oligoprogressive disease, such as in the MESO-PRIME trial [16].

Based on an innovative adjuvant radiotherapy scheme for MPM patients, we retrospectively reported our experiences using accelerated hypofractionated intensity-modulated arc therapy in tomotherapy. The aim of the treatment was palliation, and the prescription dose was 25 Gy in five fractions over five consecutive days. In considering the feasibility of an acceptable lung toxicity profile and encouraging survival rates in this retrospective analysis, a prospective monocentric pilot trial was initiated (MESO-RT). The trial’s aim was to evaluate local control and the potential for dose escalation of 30 Gy in five fractions. In the present paper, we report preliminary toxicity data as an interim analysis result of the active MESORT protocol [17].

## 2. Materials and Methods

Patient enrollment started in 2017 with the aim of enrolling 30 patients. To date, we have enrolled 29 patients of which 20 were assessed in this paper. We conducted a prospective mono-institutional clinical trial enrolling cyto-histological proven, MPM patients. The protocol inclusion criteria were defined as life expectancy greater than six months, normal organ and marrow function, and FEV1 ≥ 50. Patients after biopsy must have measurable disease defined as at least one lesion that can be accurately measured according to modified RECIST criteria; for resected patients no more than three months are allowed for RT start. Female participants of childbearing potential and male participants whose partner is of childbearing potential must be willing to ensure that they or their partner use effective contraception during the study and for four months thereafter. Ability to understand and the willingness to sign a written informed consent document is required. Exclusion criteria were: previous thorax radiotherapy, chemotherapy allowed but completed three weeks before RT starting, participation in another clinical trial with any investigational agents within 30 days before study screening, contralateral mediastinum involvement (N3) and/or M1, respiratory needing oxygen therapy, interstitial pneumopathy, active pneumonitis, fissural disease; uncontrolled intercurrent illness including, but not limited to, ongoing or active infection, symptomatic congestive heart failure. All cases were priorly discussed in a multidisciplinary lung meeting.

### 2.1. Planning Procedure and Treatment Delivery

To perform radiotherapy, patients were immobilized in the supine position with their arms overhead using a Posirest-2 (CIVCO) before the CT simulation scan. It was acquired through a 3 mm thickness slice, at times 0, 30, 60, and 150 s with free breathing. The Pinnacle treatment planning system (version 9.3) was used to contour CT simulation images. Contouring and planning procedures were reported in the previous paper [18]. All treatment plans were elaborated using IMAT (Intensity Modulated Arc Therapy) with Tomotherapy TPS (treatment planning system). Patients were treated in Tomotherapy. Image-guided radiation therapy (IGRT) was used for daily setup control. The dose prescription to the target was 30 Gy in five daily fractions (at the reference isodose 60–70%) with an internal increasing inhomogeneous dose of up to 40 Gy for GTV (gross tumor volume); the dose prescription for CTV (clinical target volume) was 30 Gy in five daily fractions without the internal increasing inhomogeneous dose if there was not a tumor positive margin [19]. In particular, the contralateral lung was the most important organ at risk and the dose constraint of V5/5 Gy proposed by Sterzing et al. [11] was respected. No specific dosimetric constraints were required for the ipsilateral lung. The organs at risk dose-volume histograms were converted to a 2-Gy equivalent dose, and we closely adhered to the dose constraints of the literature data [20], whilst maintaining the organs at risk doses below conventional fractionation values (Appendix A). Steroids (methylprednisolone 4 mg daily) were prescribed from the first day of treatment for 30 consecutive days to control radiation-induced homolateral pulmonary inflammation. The dosage was modifiable based on symptoms and indications from the patient. If patients underwent chemotherapy, radiotherapy was administered at least three weeks after the last cycle.

### 2.2. Assessing Treatment Outcome

Clinical evaluations were scheduled two, four and six months after the end of radiotherapy to record acute toxicity and then every six months thereafter for late toxicity with a time frame up until 36 months. Each follow-up visit included a patient interview, clinical examination, recent imaging, pulmonary function, and laboratory tests. In case of evidence for locoregional recurrence or distant spread, additional tests, or imaging studies, such as 18FDG PET/CT, were performed to confirm or exclude disease progression. Early chest CT scans without contrast were performed after two months, while the subsequent total-body CT scans with contrast were performed at six months and then every six months thereafter. Response and progression were evaluated using the Modified RECIST criteria for assessment of response in malignant pleural mesothelioma [21]. Pulmonary Function Tests were performed as a baseline before radiotherapy, at the first evaluation, at two months, and then before each clinical evaluation. Respiratory defects were evaluated using spirometry according to the Standardization of Spirometry proposed by the American Thoracic Society [22]. The primary endpoints of the study were to assess acute and late toxicity. The secondary objectives were overall survival (OS) for all enrolled patients and time to progression (TTP) for patients with documented progression disease. Toxicity was assessed according to the National Cancer Institute’s (NCI) Common Terminology Criteria for Adverse Events (CTCAE) v 4.03 web-based dictionary, where acute toxicity was considered from 1st RT treatment until the sixth month and late toxicity from the 6^th^ month onwards. Currently, the protocol is ongoing.

### 2.3. Statistical Analysis

The primary endpoint of this study is to evaluate acute and late pulmonary toxicity. No formal sample size calculation was done, but based on the previous study [18], it was feasible to enroll 30 patients. The enrolment period estimated was three years. For every five patients out of the first 20, toxicity monitoring was applied; we monitored the number of G3/G4 acute toxic events related to treatment after the enrolment of the aforementioned patients. The Pocock discrete toxicity boundary values were calculated considering a 10% of AE probability. The cumulative number of toxic events was compared with the prespecified boundary values. The total number of G3/4 toxicities had been equal or greater than the associated boundary value, then the study would have stopped. Considering that we did not report high-grade toxicity for the first 20 patients, enrolment is currently open and will continue until it reaches the thirtieth patient. For continuous variables, median value and minimum-maximum statistics were presented, while counts and percentages were used for categorical variables. The number and the percentage of treated patients undergoing grade 1–4 adverse events were tabulated. To evaluate FEV1, FVC, and DLCO variation from baseline value to three-months values, the Wilcoxon signed-rank test was performed. Graph boxes were used for descriptive purposes to evaluate FEV1, FVC, and DLCO over time. Overall survival time was calculated from the time of therapy start until the date of death or the date of last of the follow-up. Time to progression was calculated as the time of therapy starting until the date of progression of the disease or the date of the last assessment of the disease. Overall survival and time to progression were estimated using the Kaplan–Meier method, and the two-sided 95% confidence interval (95% CI). All statistical analysis were performed using Stata/SE version 15.1 software.

## 3. Results

In this paper, we report the results of an interim analysis of the primary objectives of the study: acute and late toxicity. All patient characteristics are shown in Table 1A,B. Between July 2017 to June 2019, 21 patients were screened and 20 were enrolled in the protocol because one patient was excluded for disease progression before starting radiotherapy. The median follow-up was 36.7 months (range: 4–45 months). In line with the literature, 80% of patients were men with a median age of 70.5 years (range 44.8–82.5), and the vast majority had undergone epithelioid histology and had a history of past asbestos exposure (76.5%). Thirty percent of patients underwent radiotherapy after undergoing a biopsy and 70% after P/D. The interval time between surgery and the beginning of radiotherapy was about three months for all patients, as defined in the trail-inclusion criteria. Standard of care Platinum-pemetrexed based chemotherapy was administered to all 20 patients with an average of four cycles (range 3–6 cycles). All patients undertook respiratory function tests to evaluate late respiratory toxicity. In total, 20 patients were considered for the evaluation of acute and late toxicity.

### 3.1. Radiation Treatments Details

All patients completed the radiotherapy schedule without interruptions. The treatment duration was five consecutive days as scheduled. The dose prescription was 30 Gy for all patients, while the inhomogeneous dose escalation to 40 Gy was prescribed to 10 patients who had the residual gross disease, which was all the six patients who had undergone a biopsy-only procedure and four patients who had undergone partial P/D. Dose distribution and dose-volume histograms (DVH) are reported in Figure 1a–c. Dose constraints were respected for all patients.

### 3.2. Acute Toxicity

No G3-G4 acute toxicity was found, while the most common acute toxicity was pneumonitis with 65.0% G1 and 10% G2. The vast majority of the G1 patients had only mild symptoms and pneumonitis was described only for radiologic features with no need for medical therapy. Four months after the end of radiotherapy, one patient developed a radiologically documented contralateral G2 pneumonia with fever, cough, and dyspnea treated with antibiotics and steroids for about three months. This manifestation was attributed to bacterial superinfection. Other respiratory toxicities were G1–G2 cough in 50% of the patients; G1 dyspnea occurred in 65% of the patients with three patients with G2 (15%). Other relevant toxicities were G1–G2 fatigue in 40%, G1 chest pain in 27%, and dysphagia in 5%. Completed toxicities are reported in Table 2.

### 3.3. Late Toxicity

No G3/G4 pneumonitis were reported. Only two (10%) cases of G1 pneumonitis were reported. Regarding cough, three patients (15%) reported G1 and two (10%) G2; G1 dyspnea occurred in 2 (10%) patients. Furthermore, we recorded G1 pericardial effusion in 10%. We observed differences in toxicity between patients who underwent radiotherapy after P/D and those treated after pleural biopsy only in terms of pneumonitis G1, cough G1, dyspnea G1 and pain G1 which were higher in the first group. One case of pericarditis was reported two years after the treatment, it was diagnosed as possible viral infection, though, we decided to consider it as late toxicity since RT was delivered to the left hemithorax. Moreover, in two female patients treated for left MPM, G1 and G2 breast disorders have been reported (Table 2).

### 3.4. Pulmonary Function Tests

The median value of Forced Expired Volume in one second (FEV1) recorded was 75.5% (range: 46–137%) before the treatment was initiated, indicating a restrictive ventilatory defect. Seventeen patients were evaluable for pulmonary function test before treatment and after three months. Baseline median value for this group was 74% (range 56–137%) underlining a worsening of a restrictive ventilatory defect even if not statistically significant (*p*-value 0.061) (Appendix A). The median value of Forced Vital Capacity (FVC) at baseline for the 20 patients was 74 (range 46–123) with progressive decreasing values through time. DLCO has also been reported with a progressive decrease over time, though, fewer measurements have been made compared to the other two measurements due to the test’s complexity and unavailability in all centers, particularly during the COVID-19 pandemic (Appendix A—Raw data).

### 3.5. Overall Survival and Time to Progression

Nine patients are still alive and undergoing follow-up: six with stable disease, while three patients with documented progression of the disease. Nine patients died during the follow-up due to disease progression. In all, these patients’ progressive disease was scored as follows: two cases of outfield peritoneal progression disease, cases of infield progression, and four patients developed contralateral and mediastinal progression disease. In particular, two patients of this group presented homolateral and contralateral lung nodules. We considered homolateral lung nodules as infield progression also if we try to spare it by high radiation doses. An important aspect to consider was that only one patient who did not undergo surgery is still alive, while all the others died within almost a year. The median OS was 33.1 months (95% CI:14.4–not estimable), though, 1-year OS was 85% (95% CI:60.4–94.9) and 3-years was 49.5% (95% CI:26.5–68.9) where the better outcome was for the patients who underwent P/D compared to patients with biopsy only; 12-months OS 92.9% (95% CI:59–98.9) and 66.7% (95% CI:19.5–90.4), respectively. More apparent distinction appears in the 36-months OS with 63.5% (95% CI:33.1–82.9) in the P/D arm with respect to 16.7% (95% CI:0.8–51.7) for the non-surgery arm (*p*-value 0.086) (Figure 2A,B). The median Time to Progression was 18.2 months (95% CI:11.3–not estimable). One- and two-year TTP were 74.4 (95% CI:48.9–88.5) and 47.8 (95% CI:24.8–67.7). Patients who underwent P/D also had a favorable median TTP compared to those who did not go through surgery (26.9 vs. 11.3 months; *p* = 0.015) (Appendix A—Raw data).

## 4. Discussion

Malignant pleural mesothelioma is a rare disease with a poor prognosis for which the role of radiotherapy (RT) is still lacking high-level evidence to guide the most effective use. In the past, the role of radiotherapy in the treatment of pleural mesothelioma was in the field of palliation to control pain or in the field of prophylaxis to prevent cutaneous metastasis in the site of previous pleural interventions. In the last years, with the advent of trimodality treatment, it has turned out that radiotherapy could have a role also in the curative setting, which can be performed in neoadjuvant but most commonly, in the adjuvant setting. The only relevant neoadjuvant study was the SMART trial (Surgery for Mesothelioma after Radiation Therapy) published by De Perrot et al. [23]. In this study, 25 Gy were delivered in five daily fractions to the entire ipsilateral hemithorax with a concomitant boost of 5 Gy to volumes at high risk, followed by Extrapleural Pneumonectomy (EPP) a week after the radiation therapy. Currently, in the most recent publications, the main role of RT treatment seems to be in the adjuvant setting, possibly after chemotherapy and surgery. The EPP approach reported relatively higher morbidity and mortality rate in addition to decreasing quality of life compared to P/D25 [24]. Furthermore, it has been reported in different studies that adjuvant RT after EPP, had important, and in some cases, fatal toxicity rates as well. The first and most crucial observation was made by Allen et al. [25], in which patients treated with IMRT with a conventional dose of 54 Gy in 28 fractions after EPP and adjuvant chemotherapy were retrospectively reviewed. The study reported a high rate of 46.1% radiotherapy-related deaths, specifically, G5 fatal pneumonitis with median onset 30 days from radiation completion, suggestive of acute toxicity. They hypothesized that the fatalities were probably a result of low dose distribution to the contralateral lung and concluded that, in addition to the V20 constraint, a V5 and mean lung dose should be considered when using intensity-modulated radiotherapy. Since then, lung-sparing surgery has become the most common surgical approach, especially for earlier stages. Because of this conservative approach, obtaining macroscopic complete resection (MCR) is less feasible with potentially lower locoregional control, therefore, a safe and efficacious RT technique had to be explored [26]. The development of new highly conformal radiotherapy techniques, such as IMRT, as described by Rosenzweig et al. [27], have enabled to optimize the delivery of high dose radiotherapy to the whole hemithorax, with acceptable pulmonary toxicity. In 2016, the breakthrough IMPRINT paper by Rimner et al. [28] determined the safety and feasibility of hemithoracic intensity-modulated pleural radiation therapy after chemotherapy and P/D, with a median PFS of 12.4 months and OS of 23.7 months. Since then, different experiences explored radiotherapy using conventional fractionation, mostly in monocentric and retrospective trials. Only a little group of studies used accelerated hypofractionated radiotherapy. One of these was our previous experience in which 36 MPM patients were treated with palliative intent using accelerated hypofractionated radiotherapy delivering from 5 to 7.5 Gy in five fractions after P/D or without any surgery [18]. In that retrospective study, no grade 4 or 5 toxicities were seen, and the median overall survival was 21.6 months for the group that received RT after P/D. Encouraged by those results, we hypothesized to increase the dose per fraction to 6 Gy per fraction for five consecutive days to improve PFS and OS without increasing pneumonitis rates. This led the way to start the prospective pilot MESO-RT trial. To our knowledge, MESO-RT is the first study that uses 30 Gy in five fractions to irradiate all the pleural volume or surgical bed of pleural volume with intact lungs. Helped by our previous experience and on account of the literature, we learned to analyze dosimetric data identifying contralateral lung dose as the major predictive factor for fatal pneumonitis. As part of the planning procedures, we fully adhered to the established IMRT constraints, particularly, the crucial constraint was the V5/5 for the contralateral lung, and as indicated by Allen et al., we used V20 and mean lung dose values of 3% and 3 Gy, respectively (Table 2). On the other hand, concerning the homolateral lung, we were able to relatively spare one-third of the intact lung volume from the dose prescription without adopting specific dose constraints. All treatments were performed with the use of IMRT, particularly IMAT in tomotherapy. This technique enabled maximum achievement in the sparing of the ipsilateral and contralateral lung, as described as well by Minatel et al. [29]. As expected, the most common and inevitable acute and late toxicity was pneumonitis, the vast majority reported only G1 radiologic with no related debilitating symptoms. In the analysis of lung dosimetric parameters, we did not find any relationship with the only patient who developed G3 pneumonitis which probably took place due to a superinfection, supported by the fact that it occurred bilaterally, the patient was treated with antibiotics and steroids for about 3 months. We observed differences in toxicity between patients who underwent radiotherapy after pleurectomy/decortication and those treated after pleural biopsy in terms of pneumonitis G1, cough G1, dyspnea G1 and pain G1 which were higher in the first group. In addition, the only significant clinical difference observed between patients with right or left disease was pericardial effusion and one case of pericarditis, which was more frequent in left-side disease. The favorable result of toxicity can depend on various aspects such as a mean lung dose lower than 17 Gy for the total lung and less than 2 Gy for the contralateral lung. From the correlation performed between the spirometric values at baseline and during follow-up, we found a progressive reduction of the three analyzed variables: Forced Expiratory Volume in the 1st second (FEV1), Forced Vital Capacity (FVC), and Diffusion Lung CO (DLCO) in the absence, however, of a relevant clinical correlation in both the acute and late interval. As a matter of fact, at the one-year evaluation, we found less evident reduction rates for all three parameters. We can deduct from this that the greatest rate of parametric reduction in respiratory function is obtained in the first year, and then stabilizes thereafter. It was relevant to note to evaluate respiratory function that 60% of patients presented with the right disease. Obviously, these data must be interpreted with caution due to the limited number of patients, due to the interim analysis of a protocol that is ongoing, and additionally, since the Pulmonary Function Tests were suspended or postponed during the COVID-19 pandemic, most patients encountered difficulties undertaking these tests in the scheduled time. We planned a strict follow-up in the first six months after RT with a clinical examination every two months. Another important aspect, as with the previous experience, was the early administration of steroids, to control acute inflammation by accelerated hypofractionation doses. The median locoregional relapse-free survival, reported in a recent systematic review [30], ranged from 12 to 16 months. In our experience, even with an interim limited cohort of 20 patients, the median Time to Progression reached 18.2 months (95% CI:11.3–not estimable) for all the patients, reinforcing the fact that RT contributes to the locoregional control. More evidently, in patients who underwent surgery, TTP reached 26.9 months, more than double the time compared to the biopsy-only cohort with 11.3 months (*p* = 0.015). Overall Survival showed encouraging results as well, median OS was 33.1 months (95% CI:14.4–not estimable), though, 1-year OS was 85 (95% CI:60.4–94.9) and 3-years was 49.5 (95% CI:26.5–68.9). Patients who underwent P/D also had a favorable median OS 45.7% (95% CI:14.4–not estimable) compared to those who weren’t amenable to surgery 15.7% (95% CI:4.1–not estimable), with 1-year and 3-years OS 92.9% (95% CI:59.1–98.9) vs. 66.7% (95% CI:19.5–90.4), 63.5% (95% CI:33.1–82.9) vs. 16.7% (95% CI:0.8–51.7), respectively, reinforcing the importance of adequate surgery for MPM patients in specialized Institute. Apart from our study, only two other prospective studies are reported, by Minatel et al. and Rimner et al. [28,29], both with more patients enrolled but with shorter follow-up and comparable results of median OS, 33 and 23.7 months, respectively. Nevertheless, our study has several limitations, for example, our cohort was non-homogeneous in terms of type and extent of surgery (pleurectomy/decortication) since patients underwent surgery in two different centers. Not all the patients were able to perform the Pulmonary Function Tests at the scheduled time due to the COVID pandemic. The same reason slowed down the patient enrollment, which to date accounts for 29 patients out of 30 expected by the trial [17].

## 5. Conclusions

Our results indicate that accelerated hypofractionated radiotherapy could play an important therapeutic role after pleurectomy/decortication or after biopsy in non-operable patients. The dose prescription, 30 Gy in five daily fractions, is a novel treatment strategy in this setting; moreover, this new fractionation appears to be safe without a high grade of lung toxicity and, as far we know, our paper is the first to report an experience of accelerated hypofractionated radiotherapy in the intact lung in malignant pleural mesothelioma with curative intent. Overall Survival and Time to Progression appear to be comparable to the conventional adjuvant scheme, though, we will have to attend the completion of the trial to calculate the definitive rates. Further analysis, ideally multicentric randomized trials with a greater number of patients, is necessary to consolidate this treatment strategy as an option for managing patients with locally advanced MPM.

## Figures and Tables

**Figure 1 cancers-15-01057-f001:**
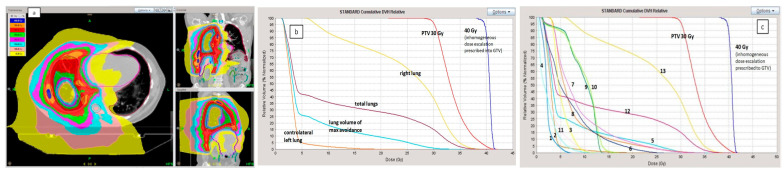
(**a**) Example of dose distribution of a patient plan treated in November 2017 for right MPM.The patient underwent 3 cycles of CHT and no surgery procedures except for biopsy. Prescription dose was 30 Gy in 5 daily fractions with inhomogeneous dose escalation up to 40 Gy inside the Gross Tumor Volume. Patient deceased 4 months after the treatment for disease complications. (**b**) dose-volume histogram in the same patient. underlining the absorbed doses by the different lung volumes. Prescribed dose was 30 Gy to the entire pleura with 40 Gy inhomogeneous dose escalation up to 40 Gy in the Gross Tumor Volume (GTV). The figure of dose distribution is reported in the paper. (**c**) dose-volume histogram in the same patient. 1: left kidney. 2: left lung (contralateral). 3: stomach. 4: bowel. 5: lung volume of max avoidance. 6: right kidney. 7: liver. 8: heart. 9: spinal cord. 10: spinal cord PRV. 11: thyroid. 12: total lungs. 13: omolateral right lung.

**Figure 2 cancers-15-01057-f002:**
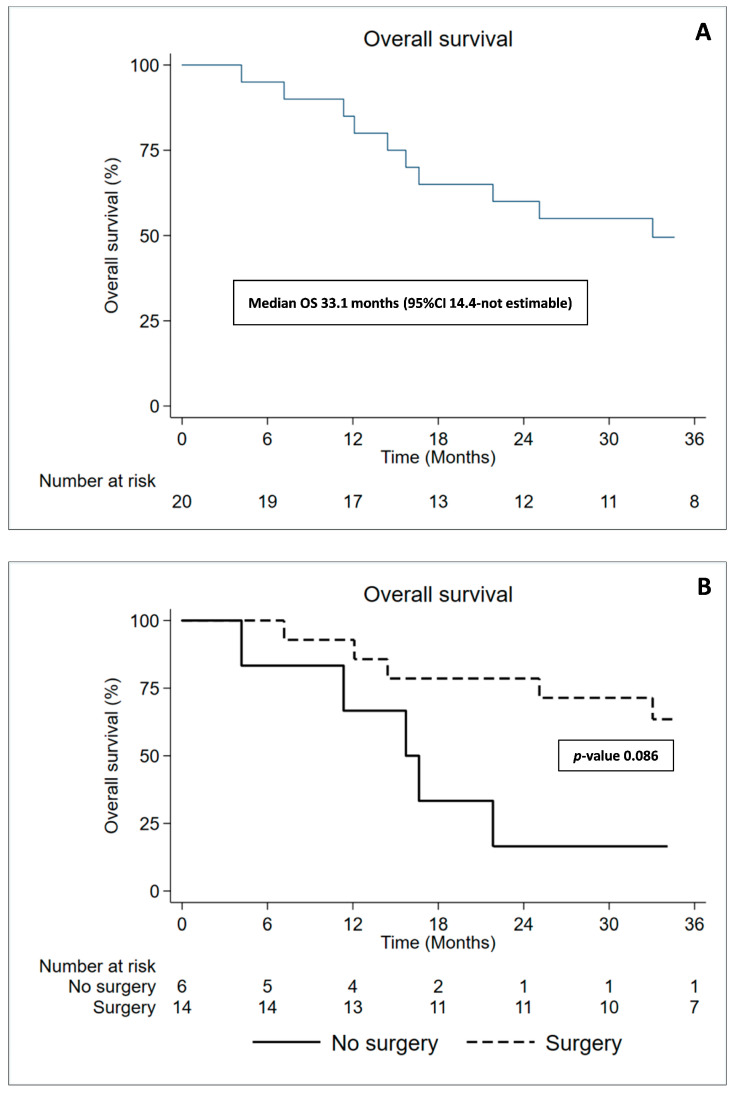
(**A**) Kaplan–Meier curves of Overall Survival for all patients; (**B**) Kaplan–Meier curves for Overall Survival difference between Surgery (P/D) and No Surgery (biopsy only). A significant difference can be seen confirming literature data regarding the importance of Surgery in the trimodality treatment.

**Table 1 cancers-15-01057-t001:** (**A**) Patients’ characteristics and (**B**) tumor characteristics.

**(A) Patients’ Characteristics**	
**Gender**	***n* (%)**
Male	16 (80%)
Female	4 (20%)
**Age**	**Years (%)**
Median (range)	72.1 (44–82.5)
**Asbestos exposure**	***n* (%)**
Yes	13 (76.5%)
No	4 (23.5%)
Unknown	3
**Smoking**	***n* (%)**
Yes	9 (64.3%)
No	5 (35.7%)
Unknown	6
**Laterality**	***n* (%)**
Right	12 (60%)
Left	8 (40%)
**(B) Tumor and Treatment Characteristics**	
**Histology**	***n* (%)**
Epithelioid	19 (95%)
Sarcomatoid	1 (5%)
**T,N Stage**	***n* (%)**
T1b	1 (5%)
T2	8 (40%)
T3	7 (35%)
T4	3 (15%)
Tx	1 (5%)
N0	14 (70%)
N1	1 (15%)
N2	3 (15%)
Nx	2 (10%)
**Previous surgery**	** *n* **
Yes	14
No	6
**Previous chemotherapy**	***n* (%)**
Pemetrexed	2 (10%)
CBCDA/Pem	6 (30%)
CDDP/Pem	11 (55%)
CDDP	1 (5%)

**Table 2 cancers-15-01057-t002:** Number of patients with reported AEs among patients with at least 1 cycle of treatment.

	Acute *n* of pts (%)			Late *n* of pts (%)		
	G1	G2	G3	G1	G2	G3
Pleural effusion	2 (10%)	2 (10%)	0 (0%)	3 (15%)	0 (0%)	0 (0%)
Pain	8 (27%)	0 (0%)	0 (0%)	2 (10%)	0 (0%)	0 (0%)
Vomiting	1 (5%)	1 (5%)	0 (0%)	1 (5%)	0 (0%)	0 (0%)
Erythema/Rush	1 (5%)	0 (0%)	0 (0%)	0 (0%)	0 (0%)	0 (0%)
Nausea	3 (15%)	0 (0%)	0 (0%)	0 (0%)	0 (0%)	0 (0%)
Fever	3 (15%)	0 (0%)	0 (0%)	0 (0%)	0 (0%)	0 (0%)
Tachycardia	1 (5%)	0 (0%)	0 (0%)	0 (0%)	0 (0%))	0 (0%)
Dyspnoea	13 (65%)	3 (15%)	0 (0%)	2 (10%)	0 (0%)	0 (0%)
Asthenia/Fatigue	5 (25%)	3 (15%)	0 (0%)	2 (10%)	1 (5%)	0 (0%)
Pneumonitis	13 (65%)	2 (10%)	0 (0%)	2 (10%)	0 (0%)	0 (0%)
Myalgia	1 (5%)	0 (0%)	0 (0%)	0 (0%)	0 (0%)	0 (0%)
Cough	7 (35%)	3 (15%)	0 (0%)	3 (15%)	2 (10%)	0 (0%)
Skin toxicity	1 (5%)	0 (0%)	0 (0%)	0 (0%)	0 (0%)	0 (0%)
Pericardial effusion	3 (15%)	0 (0%)	0 (0%)	2 (10%)	0 (0%)	0 (0%)
Loss of appetite	1 (5%)	1 (5%)	0 (0%)	0 (0%)	0 (0%)	0 (0%)
Atelectasis	1 (5%)	0 (0%)	0 (0%)	0 (0%)	0 (0%)	0 (0%)
Dyspepsia	1 (5%)	0 (0%)	0 (0%)	0 (0%)	0 (0%)	0 (0%)
Dysphagia	1 (5%)	0 (0%)	0 (0%)	0 (0%)	0 (0%)	0 (0%)
Emoftoe	1 (5%)	1 (5%)	0 (0%)	0 (0%)	0 (0%)	0 (0%)
Herpes zoster	1 (5%)	0 (0%)	0 (0%)	0 (0%)	0 (0%)	0 (0%)
Temporary disorientation	1 (5%)	0 (0%)	0 (0%)	0 (0%)	0 (0%)	0 (0%)
Asymptomatic pulmonary embolism	0 (0%)	0 (0%)	0 (0%)	0 (0%)	0 (0%)	1 (5%)
Contact dermatitis	0 (0%)	0 (0%)	0 (0%)	0 (0%)	1 (5%)	0 (0%)
Mastitis	0 (0%)	0 (0%)	0 (0%)	1 (5%)	1 (5%)	0 (0%)

## Data Availability

The datasets generated and/or analyzed during the current study are available from the corresponding author on reasonable request.

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
