# Peer review of "Malignant Pleural Mesothelioma: Preliminary Toxicity Results of Adjuvant Radiotherapy Hypofractionation in a Prospective Trial (MESO-RT)"

_cancers, 2023, doi:10.3390/cancers15041057_

Round 1

Reviewer 1 Report

In the submitted manuscript the toxicity preliminary results of the first 20 patients receiving accelerated 30 hypofractionated radiotherapy (using helical tomotherapy and intensity-modulated arc therapy) for the treatment of malignant pleural mesothelioma between July 2017 and June 2019. Patients got radiotherapy after surgical procedure (pleurectomy/ decortication or biopsy) and - if applicable - after combination chemotherapy. The prescribed dose was 30 Gy in five daily fractions, while an inhomogeneous dose escalation to 40 Gy was allowed depending on  the presence of gross residual tumor masses. Luckily, only one case of severe toxicity was reported, which was a bilateral pneumonitis that occurred two years after treatment probably due to superinfection resulting in an acceptable toxicity profile. Median time to progression lasted longer tha 1.5 years.  Overall survival rates after one resp. three years were 85% and 49.5%. Treatment of the intact lung with pleural intensity-modulated arc irradiation seemed to be safe and feasible.

However, the number of patients enrolled is quite low, updated efficacy and safety results of the next nine patients are lacking (the last recruited and considered patient was enrolled in 2019!). I am convinced the the safety results warrant further investigation of this approach, but I worry about the comparison of efficacy. Data from prospective clinical trials investigating the efficacy of pleurectomy/decortication, adjuvant or neoadjuvant chemotherapy and radiotherapy are rare and in the mentioned phase II study (IMPRINT) the percentage of patients without resection is quite higher. This might be a reason for the lower efficacy regarding OS of the IMPRINT phase II trial. 

I would appreciate if the cited literature might be reduced to the relevant publications. Particularly, articles reporting the efficacy of novel drugs which have not been approved for the treatment should not be mentioned. The citation of a guidelline published in 2010 doesn´t make any sense if a newer guideline has been published. 

Author Response

REVIEWER 1

Comments and Suggestions for Authors

In the submitted manuscript the toxicity preliminary results of the first 20 patients receiving accelerated 30 hypofractionated radiotherapy (using helical tomotherapy and intensity-modulated arc therapy) for the treatment of malignant pleural mesothelioma between July 2017 and June 2019. Patients got radiotherapy after surgical procedure (pleurectomy/ decortication or biopsy) and - if applicable - after combination chemotherapy. The prescribed dose was 30 Gy in five daily fractions, while an inhomogeneous dose escalation to 40 Gy was allowed depending on  the presence of gross residual tumor masses. Luckily, only one case of severe toxicity was reported, which was a bilateral pneumonitis that occurred two years after treatment probably due to superinfection resulting in an acceptable toxicity profile. Median time to progression lasted longer tha 1.5 years.  Overall survival rates after one resp. three years were 85% and 49.5%. Treatment of the intact lung with pleural intensity-modulated arc irradiation seemed to be safe and feasible.

However, the number of patients enrolled is quite low, updated efficacy and safety results of the next nine patients are lacking (the last recruited and considered patient was enrolled in 2019!). I am convinced the the safety results warrant further investigation of this approach, but I worry about the comparison of efficacy. Data from prospective clinical trials investigating the efficacy of pleurectomy/decortication, adjuvant or neoadjuvant chemotherapy and radiotherapy are rare and in the mentioned phase II study (IMPRINT) the percentage of patients without resection is quite higher. This might be a reason for the lower efficacy regarding OS of the IMPRINT phase II trial. 

I would appreciate if the cited literature might be reduced to the relevant publications. Particularly, articles reporting the efficacy of novel drugs which have not been approved for the treatment should not be mentioned. The citation of a guidelline published in 2010 doesn´t make any sense if a newer guideline has been published. 

AUTHOR ‘S REPLY

Dear Reviewer 1,

Thank you for all your comments and suggestions.

I’ll try to answer point by point.

  1. However, the number of patients enrolled is quite low, updated efficacy and safety results of the next nine patients are lacking (the last recruited and considered patient was enrolled in 2019!).

Reply: In the present paper, we report the results of only the first 20 patients as an interim analysis. We agree, the number of patients is quite low but this is a monoinstitutional protocol. The last recruited patient was enrolled in 2019. He was the 20th patient and allowed to start acute and late toxicity’s analysis, according statistical calculation.

  1. I am convinced the safety results warrant further investigation of this approach, but I worry about the comparison of efficacy.

Reply: We agree that safety results need further investigation of this approach. It is a particular approach and for this reason the protocol was born as monoinstitutional study in order to improve technical aspect of treatment planning. Actually, we enrolled the 29th patient. No death-related to protocol or G3/G4 toxicities were reported until now.

  1. Data from prospective clinical trials investigating the efficacy of pleurectomy/decortication, adjuvant or neoadjuvant chemotherapy and radiotherapy are rare and in the mentioned phase II study (IMPRINT) the percentage of patients without resection is quite higher.

Reply: I agree. The problem is that malignant pleural mesothelioma is a rare pathology. It means that is difficult to design a study with a large number of patients.  The ideal will be to treat them only in specialized Centres. One limitation of IMPRINT trial  was the quite higher percentage of patients  without resection in relation to the lower efficacy of OS. Probably, we need protocols for patients undergo surgery and protocols for inoperable patients.

  1. I would appreciate if the cited literature might be reduced to the relevant publications. Particularly, articles reporting the efficacy of novel drugs which have not been approved for the treatment should not be mentioned. The citation of a guideline published in 2010 doesn´t make any sense if a newer guideline has been published.

Reply: As your suggestion, I delete 4,5,6 notes of references. In the text I changed all number references, reporting new numbers in red. I also changed the note about 2010 Guidelines with 2020 Guideline. The change is reported in red colour.

Reviewer 2 Report

Dear authors,

The manuscript is interesting and well done and written. The discussion is impressive. After reviewing your study, I have several suggestion:

212: beginning of a sentence with a capital letter: "U" underlining.

229: in Table 2, I suggest presenting the data in brackets without decimal places - as shown in Table 1: 2 (10%) instead of 2 (10.0).

231-255: empty space. 

Author Response

REVIEWER 2

Comments and Suggestions for Authors

Dear authors,

The manuscript is interesting and well done and written. The discussion is impressive. After reviewing your study, I have several suggestion:

212: beginning of a sentence with a capital letter: "U" underlining.

229: in Table 2, I suggest presenting the data in brackets without decimal places - as shown in Table 1: 2 (10%) instead of 2 (10.0).

231-255: empty space. 

AUTHOR’S REPLY

Dear Reviewer 2,

Thank you for all your comments and suggestions.

  1. 212: beginning of a sentence with a capital letter: "U" underlining.

      Reply: As your suggestion, I corrected it.

  1. 229: in Table 2, I suggest presenting the data in brackets without decimal places - as shown in Table 1: 2 (10%) instead of 2 (10.0).

Reply: As your suggestion, I corrected  the Table 2 as Table 1

  1. 231-255: empty space. 

Reply: I tried to correct it.
